CTHRC1 promotes anaplastic thyroid cancer progression by upregulating the proliferation, migration, and invasion of tumor cells

Chen Yong 1 2
Jia Lanning 1
Zhao Ke 1
Chen Zuoyu 1
Han Yue 1
He Xianghui 1 hexh88@tmu.edu.cn
1 Department of General Surgery, Tianjin Medical University General Hospital , Tianjin , China
2 Department of General Surgery, Huai’an Second People’s Hospital , Huai’an, Jiangsu , China
Xu Zhijie
Electronic publication date: 2023 May 29
Publication date: 2023
Volume: 11
Electronic Location ID: e15458
Received 2022 Dec 30; Accepted 2023 May 3
Copyright: © 2023 Chen et al.
Copyright year: 2023
Copyright holder: Chen et al.
License: This is an open access article distributed under the terms of the Creative Commons Attribution License, which permits unrestricted use, distribution, reproduction and adaptation in any medium and for any purpose provided that it is properly attributed. For attribution, the original author(s), title, publication source (PeerJ) and either DOI or URL of the article must be cited.
License URL: https://creativecommons.org/licenses/by/4.0/

Keywords: Anaplastic thyroid cancer, CTHRC1, EMT, Wnt/β-catenin pathway

Funding: Tianjin Education Commission Scientific Research Project 2020KJ153 This study was funded by the Tianjin Education Commission Scientific Research Project (2020KJ153). The funders had no role in study design, data collection and analysis, decision to publish, or preparation of the manuscript.

==============================
Anaplastic thyroid carcinoma (ATC) is an extremely aggressive tumor with a high mortality rate and poor prognosis. However, the pathogenesis of ATC is complex and poorly understood, and the effective treatment options are limited. Analysis of data from the Gene Expression Omnibus (GEO) and The Cancer Genome Atlas (TCGA) databases showed that collagen triple helix repeat containing-1 (CTHRC1) was specifically upregulated in ATC tissues and was negatively correlated with overall survival (OS) in thyroid carcinoma patients. In vitro knockdown of CTHRC1 dramatically decreased the proliferation, migration, and invasion abilities of ATC cells, and in vivo studies in BALB/c nude mice confirmed that CTHRC1 knockdown significantly inhibited tumor growth. Mechanistically, CTHRC1 knockdown was found to suppress the Wnt/β-catenin pathway and epithelial-mesenchymal transition (EMT) at the protein level. These findings suggest that CTHRC1 promotes the progression of ATC via upregulating tumor cell proliferation, migration, and invasion, which may be achieved by activating the Wnt/β-catenin pathway and EMT.

Introduction

The deadliest form of thyroid cancer, anaplastic thyroid carcinoma (ATC), has a median survival rate of around 5 months, with only 20% of patients surviving an entire year (Molinaro et al., 2017). Given the disease’s aggressiveness and poor prognosis, the American Joint Committee on Cancer (AJCC) TNM system (Bible et al., 2021) categorizes all patients with ATC as stage IV. Due to the very unstable nature of the ATC genome, no curative treatments are currently available for people with the disease. The tumor mutational burden (TMB) is approximately six times greater in ATC than in differentiated thyroid cancer (DTC), and there are no ATC-specific genetic changes (Bible et al., 2021; Landa et al., 2016). Treatment with dabrafenib (BRAF inhibitor) with trametinib (MEK inhibitor) results in a surprisingly high response rate in patients with ATC harboring the BRAFV600E mutation, according to phase 2 clinical research (Subbiah et al., 2018). Since the BRAFV600E mutation is present in only about 40–70% of ATC cases (Pozdeyev et al., 2018; Xu et al., 2020), additional therapeutic targets need to be identified. Secreted extracellular matrix protein, CTHRC1, may decrease collagen matrix deposition and has been linked to tumor formation (Mei et al., 2020), which may be one such target. CTHRC1 overexpression has been observed in various solid human malignant tumors, including melanoma, gastric cancer, colorectal cancer, breast cancer, and thyroid cancer (Ip et al., 2011; Jiang et al., 2020; Kim et al., 2013; Tang et al., 2006; Wang et al., 2012). Furthermore, CTHRC1 expression has been positively associated with tumor prognosis, and it promotes invasion and metastasis by activating EMT through several signaling pathways, including the Wnt, TGF-β, and PI3K/AKT/ERK pathways (Mei et al., 2020). In papillary thyroid carcinoma (PTC), CTHRC1 knockdown suppressed TPC-1 cell proliferation and induced apoptosis (Tang et al., 2021). However, it hasn’t been known what cellular processes CTHRC1 influences to promote ATC. To address this gap, the impact of CTHRC1 knockdown on ATC proliferation, invasion, and migration was investigated in vitro and xenograft tumor formation experiments in nude mice were performed. Additionally, we also evaluated the expression of key EMT markers, β-catenin, and downstream target genes of the Wnt/β-catenin pathway in CTHRC1 knockdown and negative control groups, in order to gain insight into the possible molecular mechanism underlying the CTHRC1’s involvement in ATC. These findings may provide a foundation for future molecular-targeted therapies to inhibit CTHRC1 in ATC.

Materials and Methods

Analysis of data from online databases

The GSE33630 and GSE65144 datasets were acquired from the GEO database (https://www.ncbi.nlm.nih.gov/geo/), and data were included in a volcano plot. UALCAN (http://ualcan.path.uab.edu) was used for analyzing the expression levels of CTHRC1 in histological samples. The prognostic value of CTHRC1 expression in thyroid cancer was evaluated using the Kaplan–Meier plotter (https://kmplot.com/analysis/). P < 0.05 was considered statistically significant.

Cell lines and culture

The normal human thyroid cell line Nthy-ori3-1 was obtained from the Chinese Academy of Sciences Cell Bank (Shanghai, China). ATC tumor-derived cell lines (THJ-16T, T238, and SW1736) were presented by Antonio Di Cristofano’s Group (USA). To cultivate Nthy-ori3-1, we used an F-12K Nutrient Mixture medium, whereas the ATC cell lines were grown in RPMI 1640, both of which were supplemented with 1% penicillin-streptomycin and 10% fetal bovine serum (FBS) in a humidified incubator containing 5% CO2 at 37 °C. All the cell lines were kept at a low passage number.

Quantitative real-time polymerase chain reaction (qRT-PCR)

Total RNA was isolated from cell lines using the RNA isolator total extraction reagent (Vazyme Biotech Co., Nanjing, China). After discarding the culture medium and washing once with 1 × phosphate-buffered saline (PBS), 1 mL of the extraction reagent was added to each well of the 6-well plate. The cells were then transferred to a 1.5 mL EP tube, and lysed by repeated pulsing until fully lysed, followed by ice incubation for 5 min. Subsequently, 200 μL of chloroform was added, and the mixture was shaken vigorously for 15 s, left at 4 °C for 5 min, and then centrifuged at 12,000g for 15 min at 4 °C. The upper aqueous phase was carefully aspirated into a new EP tube, and an equal volume of pre-cooled isopropanol was added, mixed well, and left at 4 °C for 10 min. The mixture was then centrifuged at 12,000g for 10 min at 4 °C and the supernatant was discarded. Following this,1 mL of anhydrous ethanol was added, shaken well, and centrifuged at 12,000g for 5 min at 4 °C. The supernatant was discarded, and the precipitate was allowed to dry at 25 °C for 5 min before adding 30 μL of RNase-free ddH2O to dissolve the RNA precipitate. The HiScript R III 1st Strand cDNA Synthesis Kit (Vazyme, Nanjing, China) was then used to synthesize cDNA from the isolated RNA. The cDNA was diluted in RNase-free water to 5 ng/μL. SYBR qPCR Master Mix (Vazyme, Nanjing, China) and the CFX96TM Real-Time System (Bio-Rad Laboratories, Hercules, CA, USA) were used to conduct qRT-PCR, with the following steps: pre-denaturation at 95 °C for 30 s, 40 cycles of denaturation at 95 °C for 10 s, and annealing extension at 60 °C for 30 s. 2−ΔΔCT analysis was used to examine mRNA expression, and the results were standardized to β-actin levels. The specific primer sequences were as follows: CTHRC1, Forward: 5′-CAATGGCATTCCGGGTACAC-3′, Reverse: 5′-GTACACTCCGCAATTTTCCCAA-3′; β-actin, Forward: 5′-CATGTACGTTGCTATCCAGGC-3′, Reverse: 5′-CTCCTTAATGTCACGCACGAT-3′.

Western blotting

Whole cells were lysed for 2 min in RIPA buffer (R0010; Solarbio, Beijing, China) containing 1% PMSF, incubated at 4 °C for 30 min, then centrifuged at 14000 rpm at 4 °C for 10 min to obtain the protein from the supernatant. The BCA protein assay kit (PC0020; Solarbio, Beijing, China) was used for protein quantification. Equal protein samples (15 µg) were fractionated by SDS-PAGE and transferred on PVDF membranes (YA1701-1EA; Solarbio, Beijing, China). After being blocked with 5% non-fat milk for 2 h, the membranes were probed with primary antibodies against CTHRC1 (1:1000, ab256458; Abcam, Cambridge, UK), β-catenin (1:1000, A00004; Boster Bio, Wuhan, China), c-myc (1:500, BA1284-2; Boster Bio, Wuhan, China), Cyclin D1 (1:1000, BA0770-2; Boster Bio, Wuhan, China), E-Cadherin (1:1000, ab235682; Abcam, Cambridge, UK), Vimentin (1:1000, #5741; Cell Signaling Technology, Danvers, MA, USA), Snail (1:2000, A5243; ABclonal Science Inc., Woburn, MA, USA), β-actin (1:5000, AC026; ABclonal, Woburn, MA, USA), GAPDH (1:10000, TA802519BM; Origene, Rockville, MD, USA), overnight at 4 °C, after which they were subsequently incubated with the secondary antibodies (1:5000, ab150077/ab150113; Abcam, Cambridge, UK) for 40 min, at 25 °C. Finally, Liquid A and Liquid B of the ECL kit (PE0010, Solarbio, China) were mixed (1:1) and 150 μL of the ECL reagent was added to the surface of each of the protein bands, then the bands were visualized using a BLT GelView 600 Plus (BioLight, Zhuhai, China). The gray value analysis of each band was used by Image J software.

Transfection and transduction

The CTHRC1 shRNA and the negative control were purchased from Shanghai Genechem Co. (GIEL0290536; Shanghai Genechem Co., Shanghai, China). The shRNA sequence targeting human CTHRC1 was GTGAAGGAATTGGTGCTGGAT, and for the negative control, the inserted sequence was TTCTCCGAACGTGTCACGT. For lentiviral transduction, ATC cells were seeded in a 6-well plate 12 h before infection. When the cell density reached 30–40%, the culture medium was discarded, and 960 µL of complete RPMI 1640 medium, together with a lentiviral suspension at a multiplicity of infection (MOI) of 10, and 40 µL of 25× HitransG P infection enhancement solution (Genechem, Shanghai, China) were added to each well for transfection. After 12 h, the complete RPMI 1640 medium was replaced and continued to culture for 48 h, followed by puromycin selection at a dose of 2 µg/mL for 96 h, and stable transfected ATC cells were generated. Subsequent experiments were carried out using the chosen cell lines.

Cell proliferation

ATC cell proliferation was assessed using a plate colony formation assay and a Cell Counting Kit-8 (CCK-8) assay kit (40203ES60; Yeasen Biotech, Shanghai, China). For the CCK-8 assay, ATC cells were seeded at 1,000 cells per well in a 96-well plate. After cultured for 0, 24, 48, 72, and 96 h, 10 μL of CCK-8 reagent was added to each well and incubated for 2 h at 37 °C. A microplate reader (Tecan, Männedorf, Switzerland) was used to measure absorbances at 450 nm. Seeding 1,000 cells per well in 6-well plates for the colony formation assay and incubated at 37 °C with 5% CO2 for 7–10 days. The cells were then fixed with 4% paraformaldehyde (PFA), and stained with 0.1% crystal violet (CV), and the number of viable positive cells was counted manually.

Cell migration

Cell migration was assessed using Transwell migration and wound healing assays. The former was carried out in 24-well Transwell chambers (3470; Corning, Corning, NY, USA). The cell suspension (200 µL) of serum-free medium with 1.5 × 104 transfected cells was seeded into the upper chamber, and 500 µL of RPMI 1640 medium supplemented with 20% FBS was added to the lower chamber. Then the chambers were cultured in a humidified incubator containing 5% CO2 at 37 °C for 24 h. After aspirating the medium, a cotton swab was used to remove the cells that had not migrated. The migrated cells were then fixed in 4% paraformaldehyde and stained with 0.1% crystal violet for 20 min before being counted and imaged using an inverted microscope. For the wound healing assays, the transfected cells were planted in 6-well plates, and grown until 90% confluent. Linear scratches were made in the cell monolayer using a 200 µL micropipette. The cells were grown in FBS-free RPMI 1640 medium after being washed three times with PBS. The scratch widths were imaged using an inverted microscope at 0, 24, and 48 h, respectively. The wound healing percentage was analyzed by Image J software and reported as the percentage of the healing compared to the initial wound area.

Cell invasion

Cell invasion was assessed using a Transwell invasion assay in 24-well Transwell chambers, as previously described. To form a barrier, Matrigel® (BD, USA) was diluted with serum-free RPMI 1640 (1:8), and added to the upper chamber, incubated at 37 °C for gelling overnight. The remaining steps were identical to those of the Transwell migration experiment.

Animal models

Studies involving athymic nude mice were carried out following the “Guide for the Care and Use of Laboratory Animals” and sanctioned by the Tianjin Medical University General Hospital Animal Ethics Committee (approval ID: IRB2021-DW-39). Athymic nude mice were obtained from Beijing HFK Bio-Technology Co., Ltd. (China) and were raised in a pathogen-free environment with regular light/dark cycles and free access to water and food. Ten BALB/cA-nu female mice (5 weeks old) were allowed to acclimatize for a week after which they were randomly assigned to either the control or treatment group. Subcutaneous injections of CTHRC1-knockdown THJ-16T cells (5 × 106) or negative control THJ-16T cells (5 × 106) in PBS containing 10 µL Matrigel® were given to mice in the treatment and control group. The mice were observed daily, and the tumor volume and body weight were measured every 5 days. Tumor volume was determined using the method volume = length × width2/2. Once the 20 days were over, the mice were culled by carbon dioxide inhalation, and the tumors were excised. Immunohistochemistry (IHC) was performed after the paraffin sections were made.

Immunohistochemistry

Paraffin sections of 4 µm thickness were cut and dewaxed using xylene I and xylene II, hydrated in an ethanol gradient, and antigen retrieval was performed at high temperature and high pressure. Endogenous peroxidase was removed using 3% hydrogen peroxide, and the sections were then blocked with goat serum before being incubated with primary antibodies, including anti-CTHRC1 (1:500, ab256458; Abcam, UK), anti-E-Cadherin (1:100, ab235682; Abcam, Cambridge, UK), anti-Vimentin (1:200, #5741; Cell Signaling Technology, Danvers, MA, USA), anti-Snail (1:50, A5243; ABclonal Science Inc., Woburn, MA, USA) overnight at 4 °C. The sections were then incubated with a secondary antibody for 1 h at 25 °C, followed by color development using diaminobenzidine (DAB), counterstaining with hematoxylin, differentiation with hydrochloric acid ethanol, dehydration using an ethanol gradient, the sections were cleared in xylene and imaged (Nakazawa et al., 2022). All immunohistochemical stainings were evaluated blindly and independently by two investigators.

Statistical analysis

The data are presented as mean ± standard deviation. Two-tailed Student’s t-test was used to compare the two groups, while differences between multiple groups were calculated with a one-way analysis of variance. Data analysis was performed using GraphPad Prism 8, and the significance level was set at P < 0.05.

Results

CTHRC1 was upregulated in ATC tissues and cell lines

CTHRC1 expression was analyzed in ATC and normal thyroid tissues in the GEO dataset GSE33630 and GSE65144. The results demonstrated that CTHRC1 was significantly upregulated in ATC tissues (Fig. 1A). Additionally, the differential expression of CTHRC1 was compared between thyroid cancer (TC) tissues and normal thyroid tissues in the TCGA database, which further confirmed the upregulation of CTHRC1 in TC tissues (Figs. 1B and 1C). In addition, Kaplan–Meier plot analysis showed that patients with higher CTHRC1 expression in TC had significantly shorter overall survival (OS) (Fig. 1D). To investigate CTHRC1 expression further, western blotting and qRT-PCR analysis were conducted to compare the levels of CTHRC1 in the normal human thyroid epithelial cells (Nthy-ori3-1) and ATC cells (THJ-16T, T238, and SW1736). Our results showed that CTHRC1 expression was markedly increased in ATC cell lines compared to the normal thyroid cell line (Figs. 1E and 1F).

Figure 1 CTHRC1 was upregulated in ATC tissues and cell lines.

(A) The expression of CTHRC1 in ATC and normal thyroid tissues was analyzed in GSE65144 database. (B) The expression of CTHRC1 in normal thyroid and primary tumor samples based on TCGA. (C) The expression of CTHRC1 in thyroid cancer based on tumor histology from TCGA. (D) Overall survival (OS) analysis of CTHRC1 in thyroid cancer was retrieved from the Kaplan–Meier plot analysis. (E) CTHRC1 expression in ATC and normal human thyroid cell lines was analyzed by qRT-PCR. (F) CTHRC1 expression in ATC and normal human thyroid cell lines was analyzed by Western blotting. ***P < 0.001.

CTHRC1 knockdown suppressed ATC migration and invasion in vitro

The biological role of CTHRC1 in ATC in vitro was studied using THJ-16T and T238 cell lines, and the inhibitory efficacy of CTHRC1 shRNA was validated by qRT-PCR and western blotting. Transfection of the CTHRC1 shRNA significantly reduced CTHRC1 expression in ATC cells relative to the control (Figs. 2A and 2B). The effect of CTHRC1 on ATC cell migration and invasion was investigated using wound-healing and Transwell assays. The results indicated that both invasion (Fig. 2C) and migration (Figs. 2D and 3) of ATC cells were significantly reduced after CTHRC1 knockdown compared to the negative control group.

Figure 2 CTHRC1 knockdown suppressed ATC migration and invasion in vitro.

The knockdown efficiency of CTHRC1 was analyzed by (A) qRT-PCR and (B) Western blotting. (C) Transwell invasion and (D) migration assays were used to explore the effect of CTHRC1 on ATC cell invasion and migration. ***P < 0.001.

Figure 3 CTHRC1 knockdown suppressed ATC migration in vitro.

The wound-healing assay was used to explore the effect of CTHRC1 on ATC cell migration. *P < 0.05, **P < 0.01.

CTHRC1 knockdown suppressed ATC growth in vitro and in vivo

The effect of CTHRC1 on ATC cell proliferation was investigated using the CCK-8 and clonogenic assays in vitro. Our results showed that CTHRC1 knockdown significantly reduced ATC cell proliferation (Fig. 4A) and colony formation (Fig. 4B) compared to the negative control group. Moreover, a substantial decrease in tumor volume and weight were observed in CTHRC1 knockdown group compared to the control group (Figs. 4C–4F), indicating that CTHRC1 knockdown has a suppressive effect on ATC cell proliferation both in vitro and in vivo.

Figure 4 CTHRC1 knockdown suppressed ATC growth in vitro and in vivo.

(A) The Cell Counting Kit-8 and (B) colony formation assays were used to explore the effect of CTHRC1 on ATC cell proliferation. (C–E) CTHRC1 knockdown inhibited ATC tumor growth in nude mice. (F) Immunohistochemical staining was used to analyze the CTHRC1 expression in CTHRC1 knockdown and control group tumors, brown staining represented the expression of the target protein. *P < 0.05, **P < 0.01, ***P < 0.001.

CTHRC1 knockdown inhibited EMT in ATC in vitro and in vivo

EMT plays a critical role in tumor growth and is closely related to tumor cell migration and invasion (Saini et al., 2019). Therefore, we detected the changes in EMT-related molecules in ATC cells at the protein level after CTHRC1 knockdown. The result showed that the knockdown of CTHRC1 resulted in considerably lower levels of Snail and Vimentin expression than the control group, although the expression of E-Cadherin, a typical epithelial cell adhesion protein, was elevated (Fig. 5A). Furthermore, immunohistochemical staining was performed to detect E-Cadherin, Snail, and Vimentin in paraffin-embedded sections of nude mice tumor tissue sections. Interestingly, the results were consistent with those of the western blotting (Fig. 5B). These findings confirmed that the knockdown of CTHRC1 suppressed the EMT in ATC by restoring the epithelial phenotype of ATC cells.

Figure 5 CTHRC1 knockdown inhibited EMT in ATC in vitro and in vivo.

(A) Western blotting was used to analyze the E-Cadherin, Vimentin, and Snail expression in CTHRC1 knockdown and control group. (B) IHC staining was used to analyze the E-Cadherin, Vimentin, and Snail expression in CTHRC1 knockdown and control group tumors, brown staining represented the expression of the target protein. **P < 0.01, ***P < 0.001.

CTHRC1 knockdown inhibited the Wnt/β-catenin pathway

Many signaling pathways, including Notch, TGF-β, and Wnt, have a role in inducing or regulating EMT activity (Liao & Yang, 2017). Several types of cancer, including thyroid cancer, are strongly linked to the Wnt/β-catenin pathway (Martínez-Jiménez et al., 2020; Nusse & Clevers, 2017; Sastre-Perona et al., 2016). To investigate the possible molecular mechanism of CTHRC1 in ATC, the levels of β-catenin expression were examined using western blotting and found to be significantly lower in the CTHRC1 knockdown group than in the control group. Moreover, it was observed that the expression of Wnt/β-catenin downstream target genes, including Cyclin D1 and c-myc, were affected. Consistent with a decrease in β-catenin, the levels of these proteins were also reduced (Fig. 6). These results suggested that CTHRC1 knockdown may inhibit the Wnt/β-catenin pathway in ATC.

Figure 6 CTHRC1 knockdown inhibited the Wnt/β-catenin pathway.

Western blotting was used to analyze the β-catenin, Cyclin D1, and c-myc expression in CTHRC1 knockdown and control group. ***P < 0.001.

Discussion

Despite its rarity, ATC is one of the deadliest forms of malignant thyroid tumor, and its prevalence has grown in recent years (Ferrari et al., 2020; Molinaro et al., 2017). Current treatments for ATC, including surgery, chemotherapy, and radiation, have not been shown to significantly slow the disease’s progression or improve patients’ prognoses (Bible et al., 2021; Perrier, Brierley & Tuttle, 2018). Therefore, new therapeutic techniques are desperately required, and targeted treatments, which attempt to limit the activation of oncogenes in ATC cells, have emerged as the most popular treatment modalities now under development. Consequently, it is crucial to clarify the underlying molecular pathways that promote ATC development to pinpoint prospective treatment targets.

The oncogene CTHRC1 is critical for developing many types of solid tumors (Chen et al., 2013; Ding et al., 2020; Park et al., 2013; Sial et al., 2021). To examine the in-vitro biological roles of CTHRC1 in ATC, cell lines with stable CTHRC1 knockdown were established. Knockdown of CTHRC1 slowed the proliferation, migration, and invasion of ATC cells, indicating a function for CTHRC1 in the development of ATC. However, CTHRC1 knockdown did not affect apoptosis in ATC cells. While prior studies showed that CTHRC1 knockdown suppressed the proliferation and induced apoptosis in PTC cells (Tang et al., 2021). In general, the inhibition of tumor cell proliferation is accompanied by the promotion of apoptosis (Wang et al., 2020). But our results are interesting, and perhaps this is why the pathogenesis of ATC is complex and difficult to treat. On the other hand, prior research indicated that CTHRC1 enhanced cell adhesion, invasion, and metastasis but had no discernible impact on cell proliferation in gastric cancer (Ding et al., 2020), hepatocellular carcinoma (Chen et al., 2013), and pancreatic cancer (Park et al., 2013). These results imply a multifaceted function for CTHRC1 in malignancies. Nevertheless, our study indicates that silencing CTHRC1 can robustly suppress tumor growth in vitro and in vivo, targeting CTHRC1 may be an effective strategy to treat ATC.

Enhanced cell migration is mainly responsible for the increased invasiveness of tumor cells. Cancer cells release CTHRC1 into their microenvironment, enhancing their motility and the motility of neighboring cancer cells by activating RhoA signaling (Chen et al., 2013). Compared to normal thyroid tissue and differentiated thyroid cancer, the upregulation of Wnt signaling in ATC results in a dedifferentiated phenotype and confers metastatic properties on ATC (Liu et al., 2018). The present study revealed that CTHRC1 knockdown resulted in a significant decrease in β-catenin, Cyclin D1, and c-myc expression compared to the control group, indicating that CTHRC1 knockdown may suppress the Wnt/β-catenin signaling pathway, leading to inhibition of ATC cells proliferation, invasion, and migration. In fact, the most intuitive was to clarify the decrease in β-catenin translocation to the nucleus with CTHRC1 knockdown. Thus, it could be proved that the Wnt/β-catenin pathway was inhibited. The nuclear translocation of β-catenin requires further confirmation through immunofluorescence assays. Additionally, future studies will include overexpression of CTHRC1 to investigate changes in the Wnt/β-catenin pathway to confirm that CTHRC1 promotes ATC progression by activating this pathway.

The EMT in epithelial cells results in their dedifferentiation and loss of epithelial traits before re-differentiation and the acquisition of mesenchymal characteristics. Enhancement of tumor metastasis, migration, invasion, and resistance to apoptosis have all been associated with EMT (Lamouille, Xu & Derynck, 2014; Pastushenko & Blanpain, 2019; Ren et al., 2018). EMT is mainly responsible for cancer cell migration and invasion (Yilmaz & Christofori, 2009). In cancer, the Wnt/β-catenin signaling pathway is a classic route that activates EMT (Bugter, Fenderico & Maurice, 2021; Zhang et al., 2021; Zhang & Wang, 2020). Furthermore, β-catenin may control the EMT-related transcription factors expression as the primary downstream effector of Wnt/β-catenin signaling (Gonzalez & Medici, 2014). The correlation between CTHRC1 and E-Cadherin and Vimentin expression in PTC tissues suggested that CTHRC1 may be involved in the EMT in PTC (Tang et al., 2018). Combined with previous research, we speculate that CTHRC1 may induce the EMT in ATC by activating the Wnt/β-catenin pathway; however, further experiments are still needed to verify our hypothesis.

Conclusions

Our investigation has revealed that CTHRC1 plays a critical role in promoting the progression of ATC by upregulating the proliferation, migration, and invasion of tumor cells. This is likely accomplished through the activation of the Wnt/β-catenin pathway and EMT. These findings provide additional theoretical support for understanding the mechanism underlying the development of ATC and may have significant implications for identifying potential therapeutic targets.

Supplemental Information

Supplemental Information 1 Raw data of Figure 1.

Click here for additional data file.

Supplemental Information 2 Raw data of Figure 2.

Click here for additional data file.

Supplemental Information 3 Raw data of Figure 3.

Click here for additional data file.

Supplemental Information 4 Raw data of Figure 4.

Click here for additional data file.

Supplemental Information 5 Raw data of Figure 5.

Click here for additional data file.

Supplemental Information 6 Raw data of Figure 6.

Click here for additional data file.

Supplemental Information 7 CTHRC1 knockdown did not affect apoptosis in ATC cells.

Click here for additional data file.

Supplemental Information 8 The differences in genetic alterations between the 3 ATC cell lines.

Click here for additional data file.

Supplemental Information 9 Report of cell line identification.

Click here for additional data file.

Supplemental Information 10 Author Checklist.

Click here for additional data file.

Additional Information and Declarations

Competing Interests

Author Contributions

Animal Ethics

Data Availability

The authors declare that they have no competing interests.

Yong Chen conceived and designed the experiments, performed the experiments, authored or reviewed drafts of the article, and approved the final draft.

Lanning Jia performed the experiments, authored or reviewed drafts of the article, and approved the final draft.

Ke Zhao analyzed the data, prepared figures and/or tables, and approved the final draft.

Zuoyu Chen performed the experiments, prepared figures and/or tables, and approved the final draft.

Yue Han performed the experiments, prepared figures and/or tables, and approved the final draft.

Xianghui He conceived and designed the experiments, authored or reviewed drafts of the article, and approved the final draft.

The following information was supplied relating to ethical approvals (i.e., approving body and any reference numbers):

The study was sanctioned by the Tianjin Medical University General Hospital Animal Ethics Committee (IRB2021-DW-39).

The following information was supplied regarding data availability:

The raw measurements are available in the Supplemental Files.

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
