# Peer review of "CTHRC1 promotes anaplastic thyroid cancer progression by upregulating the proliferation, migration, and invasion of tumor cells"

_PeerJ, doi:10.7717/peerj.15458_

## Round 0.1 · original submission · Major Revisions

The reviewers suggested the key comments, in order to improve the quality.

·

Basic reporting

This manuscript by Chen and coworkers titled ‘CTHRC1 promotes anaplastic thyroid cancer progression via the Wnt/β-catenin signaling pathway’ describes the effect of CTHRC1 on cancer phenotypes in thyroid cancer progression. This is an excellent study, and the results were described well and adequately supported by the figures. Introduction is written well and largely relevant to the context.
I ask authors to address the following concerns:
1. In lines 42-50, you gave an excellent description of what has been known of CTHRC1 in cancer development. But you ended this paragraph stating that mechanism of CTHRC1 is carcinogenesis is not known. It is not wrong to say that. But it is not relevant for the data shown in the manuscript. You did not establish the mechanism. There is some preliminary evidence, but it is not good enough to conclusively prove what you claimed. However, the data you presented in the manuscript is adequate to say the CTHRC1 influences several hallmark phenotypes of thyroid cancer cell. Therefore, I suggest you replace the lines 48 – 50 with sentences that convey that it hasn’t been known what cellular processes CTHRC1 influences to promote thyroid cancer. I feel that would improve the flow of information in introduction and specifically highlight what this manuscript is addressing.
2. In continuation with the above concern, there is no conclusive evidence in the manuscript to claim that CTHRC1 uses WNT/beta catenin signaling to drive thyroid cancer progression. The only evidence is downregulation of beta catenin following knockdown of CTHRC1. This does not establish that CTHRC1 uses WNT signaling. The claims in this aspect were overstated and not supported by the data. Therefore, I suggest the authors to do one of the following two:
A) Please remove sentences claiming that CTHRC1 promoted cancer through Wnt/beta-catenin signaling. This would also warrant a change in title of the manuscript as well.
B) Please show that beta catenin translocation into nucleus decreased after knockdown of CTHRC1 and it was increased following reintroduction of CTHRC1 into the cell by ectopic expression. In addition, please show that mRNA levels of exclusive transcriptional targets of nuclear beta catenin (eg. AXIN2) changed depending on altering nuclear beta catenin levels in the cells in this context. Lastly, please show that the phenotypic changes detected after CTHRC1 knockdown could be reversed when effectors of wnt/beta catenin signaling.

Experimental design

1. Can you improve the contrast of the photographs shown in Figure panel 2C? Migrating front is not seen clearly in the current micrograph.
2. Can you replace mice photograph in Figure 3C? It is not clear where the tumor is on the mice. Since the experiment was already performed, if authors do not have a different and clear picture, I suggest then to remove the photo. The excised tumor arranged in the order are good enough for describing the data. However, if you chose to replace the mice photograph, please make sure that tumors and their size is appreciable in the photograph.

Validity of the findings

1. Lines 69: What is the extraction reagent? If you can elaborate RNA isolation procedure, it will allow others to reproduce your experiments.
2. Lines 83-85: What did you do with secreted proteins? Which experiments shown in the manuscript were performed on secreted proteins?
3. Lines 150-155: Please elaborate this method. Please explain it step by step. Were the slides deparaffinized twice? Why?
4. Figure 1 legend: Please write Student’s t test.

Reviewer 2 ·

Basic reporting

1) Exciting study but the authors need to work hard to improve the writing of the manuscript. Plenty of grammatical and structural errors were detected. Suggest going for a proofreading service.

2) Title can be better. CTHRC1 activates or inhibits the Wnt/beta-catenin pathway?

3) Abstract is needed to be improved. Kindly correct the grammatical and structural errors. Please be specific on the activation or inhibition on the EMT and Wnt/beta-catenin pathway.

4) Sufficient background but it will be better if include those updated articles (example: DOI 10.12122/j.issn.1673-4254.2021.04.10)

5) Make sure to describe the abbreviations first before using them.

6) Raw data is shared but it is not complete.

Experimental design

1) Line 68-71: Kindly provide the RNA purity data upon extraction or after cDNA synthesis.

2) Line 72-78: Please include the thermal cycle conditions for the reproducibility of data

3) Line 82-85: It is impossible to collect secreted protein (soluble) by using centrifugation. Please correct this.

4) Line 82-85: How do the authors standardise the collection of secreted protein and for loading? No other secreted protein was used to serve as a control. This is important to demonstrate the changes of expression are due to treatment/groups but not due to the collection, procedure or decrease in secretion.

5) Please include the catalogue number for reagents like PVDF, secondary antibodies, ECL reagent, shRNA, CCK-8, etc.

6) Some catalogue numbers for primary antibodies are not correct. Do not confuse it with strain. Please check.

7) Line 101-103: The authors should not mention the sense or antisense strands of shRNA. Please report the correct complementary sequence.

8) Line 101-104: This is one of the major concerns. The negative control (control shRNA) in the vector is not mentioned. Without this, we are not the obtained results are due to knockout of CTHRC1 or due to viral infection.

9) Line 104-106: The details of the procedure are needed. Plate the cells in 6-well or 12-well? What is the cell density or confluency? Did the authors use OPTI-MEM or polybrene?

10) Line 111: What is the cell density/seeding concentration for transfected cells for CCK assay?

11) Line 117-131: The authors did not mention it clearly on the use of the transwell system. The catalogue number is not stated, so I assume it was a Boyden chamber. The authors need to describe how much of cells were seeded, and then how the authors identify migrated and invasive cells.

12) Line 141: it should be wild-type, not untreated.

13) Line 147, 155: Do the authors quantify the IHC signal via software (like ImageJ)?

14) Line 155: state the catalogue number for the secondary antibody and the dilution.

15) Line 158: For multiple group comparison, ANOVA is preferable for normally distributed data with a maintained 5% of type 1 error. Any justification for not using it? Did the authors use an adjusted p-value for multiple comparisons?

Validity of the findings

1) How did the authors detect the blot? The use of gel-doc or x-ray film method is not mentioned in the methodology.

2) Figure 1B, 2B, 4A and 4B: Please include all the raw data for blot (at least 3 sets) and attach the calculation for relative expression.

3) Figure 2A: The raw data is confusing: A)Tthere are THJ16T 6-1, THJ16T 8-1, T238 6-1 and T238 8-1. B) The calculated is not matched with the reported value.

4) Figure 2C, D and E: Data data is not included.

5) Some of the quality of images are very poor and low in resolution. Please use images with good resolution.

6) How do the authors quantify the protein bands and calculate the relative expression? It was not mentioned in the methodology.

7) How do the authors quantify the wound healing percentage? It was not mentioned in the methodology.

8) Figure 3F: Kindly label the histological/pathological features

9) Figure 4A, 4B: Ideally the authors should check the CTHRC1 status to ensure the successful and sustained knockdown of the CTHRC1 protein.

10) All the proteins of interest shall be mentioned with their respective kDa.

11) Kindly include the body weight of the mice after xenografting.

12) Standardise the naming and avoid typing errors like THJ16T in Figure 2A, Wound healing percentagr for Figure 2C.

Additional comments

1) From the current data, there is a decrease in beta-catenin level. However, this finding is still inconclusive by relying on single protein (beta-catenin) in Wnt/beta-catenin pathway. Beta-catenin is possibly to be degraded by caspases during cell death processes (PMID: 26040106). To rule this possibility out, the authors need to check the expression of cleaved caspase-3 and include it in Figure 4.

Reviewer 3 ·

Basic reporting

In this manuscript the authors have investigated the involvement of CTHRC1 in the in-vitro proliferation, migration, and invasion of ATC cells. The authors describe some interesting in vivo mice work but the rest of the manuscript is generally underdeveloped and requires substantial work. In particular, the authors will need to address the following:

Major comments
1. The quantity of text and level of description provided in most sections is generally insufficient. In the introduction and discussion, for instance, it would be helpful if the authors included recent literature in which CTHRC1 has been studied in thyroid cancer. Examples include: (1) Z Tang, X Ding, S Qin, C Zhang. Nan Fang Yi Ke Da Xue Xue Bao. Effects of RNA interference of CTHRC1 on proliferation and apoptosis of thyroid papillary cancer TCP-1 cells in vitro. 2021 Apr 20;41(4):549-554. doi: 10.12122/j.issn.1673-4254.2021.04.10; (2) Z N Tang, Y C Wang, X X Liu, Q L Liu. An immunohistochemical study of CTHRC1, Vimentin, E-cadherin expression in papillary thyroid carcinoma. Lin Chung Er Bi Yan Hou Tou Jing Wai Ke Za Zhi. 2018 Apr;32(8):595-598. doi: 10.13201/j.issn.1001-1781.2018.08.009.
2. The authors have not included any clinical-related data to support their work. At the very least the authors should include thyroid TCGA data on CTHRC1 expression, and the relationship to patient survival (i.e. high versus low CTHRC1 expression) as a point of reference. The inclusion of CTHRC1 expression data from anaplastic thyroid cancer datasets available at the GEO repository (https://www.ncbi.nlm.nih.gov/gds/) would also improve the impact of the manuscript.
3. Please include recent STR profiles of all thyroid cell lines used in the study. A table outlining the main differences between the 3 anaplastic thyroid cancer cell lines (i.e. genetic alterations) would also be helpful for a general audience.
4. Figures 2D and 2E: Please include statistical analysis for these datasets, as well as scale bars.
5. Discussion: Please comment further on any possible approaches to target CTHRC1 and downstream pathways as a therapeutic strategy for ATC.
6. Figure 4: Please include IHC images of CTHRC1-deficient mice tumors using markers of EMT and Wnt/-catenin signaling to validate the in vitro findings.
7. Figure 3B: Please include statistical analysis for this dataset.
8. Please can you replace Western blot images of poor quality (e.g. Figure 4A: E-cadherin; Figure 4B: beta-catenin).

Minor comments
1. Please be consistent in the use of “h” versus “hours” in the methods section.
2. Line 145, please use the term “culled” instead of “put to death”.
3. Please include molecular weight markers on all western blots.
4. Figure legends: Please correct the error with the “<” symbols related to the P-values.
5. Please add colour to graphs to help improve the overall quality of presentation.
6. The font size used in many figures is too small. Please can you increase the font size where appropriate or the relative size of the figure panels.

Experimental design

No comment

Validity of the findings

No comment

Additional comments

No comment

---

## Round 0.2 · Minor Revisions

The reviewers have suggested several minor comments.

·

Basic reporting

Authors addressed my concerns. I have no further comments.

Experimental design

-

Validity of the findings

-

Reviewer 2 ·

Basic reporting

Comment 1. Exciting study but the authors need to work hard to improve the writing of the manuscript. Plenty of grammatical and structural errors were detected. Suggest going for a proofreading service.
Response 1 Thank you for your good suggestions. We have engaged a native English speaker to proofread the manuscript and correct grammatical and structural errors.

Follow-up comment: Several grammatical errors are still detected. Please proofread again. Try checking this with track-changes of Simple Markup.


Comment 2. Title can be better. CTHRC1 activates or inhibits the Wnt/beta-catenin pathway?
Response 2 Thanks for your good suggestions. CTHRC1 may activate the Wnt/β-catenin pathway, but some experimental evidence is still missing. We will continue our research to confirm that CTHRC1 activates the Wnt/β-catenin pathway in ATC later. So we have changed the title of the manuscript as CTHRC1 promotes anaplastic thyroid cancer progression by regulating proliferation, migration and invasion of tumor cells

Follow-up comment: Avoid using ambiguous terms like regulate, affect... Suggest: "CTHRC1 promotes anaplastic thyroid cancer progression by upregulating proliferation, migration and invasion of tumor cells"

Experimental design

Comment 1 Line 68-71: Kindly provide the RNA purity data upon extraction or after cDNA synthesis.
Response 1 The mRNA concentration we measured was between 400-900 ng/μL, and after synthesized cDNA, which was diluted to 5 ng/μL with RNase-free water. In addition, we have added it in lines 94-95.
Line 94-95: The cDNA was diluted in RNase-free water to 5 ng/μL.

Follow-up comments: The authors did not address my comment. Kindly provide the RNA purity data.


Comment 3 Line 82-85: It is impossible to collect secreted protein (soluble) by using centrifugation. Please correct this.
Response 3 Thanks for your good suggestions. Ultrafiltration, precipitation and dialysis are the three main methods used to extract secreted proteins. (Cao J, Shen C, Zhang J, Yao J, Shen H, Liu Y, Lu H, Yang P. Comparison of alternative extraction methods for secretome profiling in human hepatocellular carcinoma cells. Sci China Life Sci. 2011 Jan;54(1):34-8. doi: 10.1007/s11427-010-4122-1. Epub 2011 Jan 21. PMID: 21253868). We obtained the secreted protein by ultrafiltration concentration through ultrafiltration tubes (10K MWCO, Amicon Ultra-15; Sigma-Aldrich, St. Louis, MO, USA), and centrifuged at 4500 g at 4 °C for 50 minutes to isolate the secreted protein.

Follow-up comments: (1) The authors have to be clear on “secretome”, “secretory protein” and the current term “secreted protein”. Some secreted proteins will be found within the extracellular vesicle or exosome, then centrifugation method is fine to concentrate them. But if CTHRC1 is a soluble secreted protein, you need to “precipitate” the protein to make it partially or insoluble during the centrifugation process (DOI: 10.3389/fncel.2016.00070). From my experience, the authors were collecting secretome, exosome, cellular debris and apoptotic bodies, but not soluble secreted proteins. To support this, as shown in your Fig 1F, there is beta-actin detection where beta-actin is a non-secreted structural protein. The presence of beta-actin means, either there is cellular contamination or from secretome or apoptotic bodies. Centrifuge at 4500 g for 50min may not be able to fully collect all secretome or exosome. So the authors might just collect cellular debris or some secretomes.
(2) How did the authors collect those secreted proteins? Supernatant or pellet? Did the authors perform washing on the collected protein?
(3) Your current centrifugation setting is different from the referred article (DOI: 10.1007/s11427-010-4122-1). Any justification for this?


Comment 4 Line 82-85: How do the authors standardise the collection of secreted protein and for loading? No other secreted protein was used to serve as a control. This is important to demonstrate the changes of expression are due to
treatment/groups but not due to the collection, procedure or decrease in secretion.
Response 4 We are thankful to the reviewer for pointing out this question. It is absolutely true that β-actin is not a secreted protein. Since we did not get any internal control over secreted proteins, we normalized by taking an equal-quality protein sample after the secreted protein concentration was measured. In addition, according to the relevant literature (Li J, Wang Y, Ma M, Jiang S, Zhang X, Zhang Y, Yang X, Xu C, Tian G, Li Q, Wang Y, Zhu L, Nie H, Feng M, Xia Q, Gu J, Xu Q, Zhang Z. Autocrine CTHRC1 activates hepatic stellate cells and promotes liver fibrosis by activating TGF-β signaling. EBioMedicine. 2019 Feb; 40:43-55. doi: 10.1016/j.ebiom.2019.01.009. Epub 2019 Jan 11. PMID: 30639416; PMCID: PMC6412555. Ding X, Huang R, Zhong Y, Cui N, Wang Y, Weng J, Chen L, Zang M. CTHRC1 promotes gastric cancer metastasis via HIF-1α/CXCR4 signaling pathway. Biomed Pharmacother. 2020 Mar; 123:109742. doi: 10.1016/j.biopha.2019.109742. Epub 2019 Dec 25. PMID: 31855733), they extracted the whole protein to detect the expression of CTHRC1, and used the GAPDH to normalize for protein loading. We also performed the same experiment and used the β-actin to normalize for protein loading. The results showed that the difference in CTHRC1 expression between the normal thyroid cell and ATC cells was consistent with the difference in CTHRC1 in secreted proteins. In the manuscript, we describe the ultrafiltration used to extract the secreted proteins, but unfortunately, no other secreted protein served as a control. At the same time, we took the same approach as in the previous literature to perform the experiments and analysis, so we apologize for any confusion caused to the reader. Thank you once again for the opportunity to clarify.

Follow-up comments: To make it right, kindly do not confuse with the above 2 articles as they overexpressed and collected the intracellular CTHRC1, so a beta-actin/GAPDH are suitable for use. In your study, a total protein staining like Ponceau S staining should be used and have to quantify the whole lane. Another alternative is using stain-free imaging from Bio-rad or other manufacturers.


Comment 9 Line 104-106: The details of the procedure are needed. Plate the cells in 6-well or 12-well? What is the cell density or confluency? Did the authors use OPTI-MEM or polybrene?
Response 9 Thanks for your good suggestions. The section on our cellular lentivirus infection experiments has been amended to describe the experimental steps in detail in lines 128-138.
Lines 128-138: The CTHRC1 shRNA and the negative control were purchased from Shanghai Genechem Co (GIEL0290536, China). The shRNA sequence targeting human CTHRC1 is GTGAAGGAATTGGTGCTGGAT, for the negative control, the inserted sequence is TTCTCCGAACGTGTCACGT. For lentiviral transduction, ATC cells were seeded in a 6-well plate 12h before infection, when the cell density reached 30-40%, discarded the culture medium, then 960 µL complete RPMI 1640 medium, a multiplicity of infection (MOI) of 10 lentiviral suspension and 40µL 25× HitransG P infection enhancement solution (Genechem, China) were added to each well for transfection. After 12 h infection, the complete RPMI 1640 medium was replaced and continued to culture for 48 h, followed by puromycin selection at a dose of 2 µg/mL for 96 h, and stable transfected cells were generated. Subsequent experiments were carried out using the chosen cell lines.

Follow-up comments: Complete media is not recommended to be used for transfection. Kindly confirm this before I comment further.


Comment 13 Line 147, 155: Do the authors quantify the IHC signal via software (like ImageJ)?
Response 13 We used the following judging criteria. The intensity of immunohistochemistry staining was scored as 0 (negative), 1 (weakly positive), 2 (moderately positive), or 3 (strongly positive). The staining extent was scored as 0 (0), 1 (≤10%), 2 (10-50%), 3 (50%-80%), and 4 (>80%), and the final IHC score was calculated by multiplying the staining intensity score and positive staining percentage score, We compared the scores of the CTHRC1 knockdown group and the negative control group.

Follow-up comments: Blinding is important for manual scoring. Kindly mention whether you applied experimental blinding in this scoring.


Comment 15 Line 158: For multiple group comparison, ANOVA is preferable for
normally distributed data with a maintained 5% of type 1 error. Any
justification for not using it? Did the authors use an adjusted p-value for
multiple comparisons?
Response 15 Thanks for your good suggestions. In order to compare CTHRC1 expression in anaplastic thyroid cancer and normal thyroid cell lines, we used different ATC cells compared to the normal group, respectively. We did not make two-by-two comparisons between multiple groups, the two-tailed Student’s t-test evaluated any significant differences between the two groups. The reason why we used Student’s t-test is that we only did comparisons between two groups and no multiple-group comparisons were made.

Follow-up comments: The authors did not address or understand my comments. By taking Fig 1C as an example, where the authors ran 4 student’s T-tests and the type 1 error from that analysis is now (4 x 5%). The authors shall run ANOVA and only focus on the comparison with normal and the type 1 error will remain at 5%. Kindly repeat this with ANOVA

Validity of the findings

Comment 8 Figure 3F: Kindly label the histological/pathological features
Response 8 In Fig. 3F, we intended to express that the expression of CTHRC1 in tumor-forming tissues of nude mice in the CTHRC1 knockdown group is lower than in the negative control group.

Follow-up comments: The authors did not address my comment. At least mention in the legend the indication of brown color. This is good for some readers that new in the field.


Comment 9 Figure 4A, 4B: Ideally the authors should check the CTHRC1 status to ensure the successful and sustained knockdown of the CTHRC1 protein.
Response 9 Thanks for your good suggestions. We totally agree with you. We confirmed the knockdown of CTHRC1 after each protein extraction and then performed the relevant Western-blot experiments within two weeks.

Follow-up comments: The authors haven’t attach the relevant data. Figure 2?


Comment 11 Kindly include the body weight of the mice after xenografting.
Response 11 The weight of the mice did not change much, and the mental state was good. We have added it to the manuscript.

Follow-up comments: The authors haven’t attach the relevant data.

Additional comments

Comment 1 From the current data, there is a decrease in beta-catenin level. However,
this finding is still inconclusive by relying on single protein (beta-catenin) in
Wnt/beta-catenin pathway. Beta-catenin is possibly to be degraded by
caspases during cell death processes (PMID: 26040106). To rule this
possibility out, the authors need to check the expression of cleaved caspase-3
and include it in Figure 4.
Response 1 Thank you for your comments and suggestions and agree with your point of view. In our study, we also measured the expression of cyclin D1 and c-myc, which are downstream target genes of the Wnt/β-catenin pathway. Consistent with a decrease in β-catenin, these molecule expression levels also decreased after CTHRC1 knockdown. In addition, we performed flow cytometry to examine the effect of CTHRC1 knockdown on ATC apoptosis. The results showed that CTHRC1 knockdown did not affect apoptosis, which suggested that the decrease of β-catenin was not caused by cell death. Therefore, we may not need to detect the expression of cleaved caspase-3.

Follow-up comments: Thank you the authors. Kindly include this as supplementary data and in the discussion as well. However, be careful when discussing this as the knockdown of CTHRC1 did not cause any ATC apoptosis. This may challenge its essential oncogenic role in ATC.

Reviewer 3 ·

Basic reporting

Overall, the authors have addressed the majority of my concerns and the manuscript is much stronger with greater impact. Some minor concerns are:
1. The references need to be in alphabetical order.
2. Line 76. The authors indicate that the cell lines were presented to them. Do they mean the cell lines were provided to them? Please clarify.
3. Please provide a table outlining the main differences between the 3 anaplastic thyroid cancer cell lines (i.e. genetic alterations, sources etc) which would be helpful for a general audience.
4. The authors indicate that the cell lines were not contaminated. Did they check their cells for mycoplasma contamination? It would be helpful if the authors included a sentence in the methods section stating that they checked their cells for mycoplasma contamination and kept them at low passage number.
5. If the authors have not checked their cell lines by STR profiling then they should state in the methods section that the authenticity of their cell lines was not verified. These are mandatory checks for any study performed using cell lines.

Experimental design

N/A

Validity of the findings

N/A

Additional comments

N/A

---

## Round 0.3 · accepted · Accept

The comments have been well-addressed.